# Flexural Edge Waves in a Thick Piezoelectric Film Resting on a Winkler Foundation

**Saad Althobaiti [1],\* and Muhammad A. Hawwa [2]**

1    Department of Sciences and Technology, Ranyah University College, Taif University, P.O. Box 11099, Taif 21944, Saudi Arabia
2    IR Center of Advanced Materials, Mechanical Engineering Department, King Fahd University of Petroleum & Minerals, Dhahran 31261, Saudi Arabia; drmafh@kfupm.edu.sa
\*    Correspondence: snthobaiti@tu.edu.sa

**Abstract:** This paper is concerned with the analysis of bending edge waves travelling in a thick Kirchhoff-type piezoelectric film resting on a Winkler–Fuss foundation. The electromechanical coupling is accounted for in defining flexural rigidities to embrace piezoelectric and dielectric material constants. Dynamics is solved analytically and demonstrated numerically by assuming harmonic wave propagation. It is observed that an increase of the voltage leads to a decrease of the critical velocity, while an increase of the Winkler's constant leads to an increase of the critical velocity.

**Keywords:** flexural edge wave; thick piezoelectric plate; Winkler foundation

## 1. Introduction

Wave propagation in piezoelectric plates can be traced back to the work of Tiersten [1], who studied the effect of electromechanical coupling on wave dispersion characteristics, showing that three solutions can be obtained for a given frequency. Schmidt [2] and Syngellakis and Lee [3] derived a dispersion relationship for plane waves propagating in an infinite plate using a 3D linear theory of piezoelectricity. Ramos and Otero [4] identified four propagating modes in a piezoelectric slab with hexagonal symmetry and compared their numerical results with experimental ones. Jin et al. [5] studied the propagation of Lamb waves in a piezoelectric layer set down on a metallic half space, showing that the dispersion curves converged to the transverse mode of the piezoelectric layer at high wavenumbers. Lee and Liu [6] investigated the effects of mechanical damping and current conduction resistance on the dispersion curves that were computed for real frequencies and complex wave numbers. Mezheritsky and Mezheritsky [7] investigated the effects of energy loss on the characteristics plane waves propagating in a dissipative piezoelectric plate. They also calculated complex wavenumbers corresponding to real frequencies. Wu et al. [8] derived approximate equations for dispersion modes in a piezoelectric plate under thickness poling and showed that mode coupling could not be neglected if the plate was relatively thick.

In the above-cited publications, authors have studied wave propagation in infinitely extended piezoelectric plates. However, when considering piezoelectric plates utilized in smart-material applications (such as piezoelectric transducers or surface acoustic wave devices), one finds that it is necessary to study piezoelectric waveguides with various boundary conditions. It is widely observed that piezoelectric plates are normally attached to supporting/hosting structures. In interesting applications, one can find piezoelectric thick films deposited on elastic compliant substrates (e.g., [9,10]). Of special interest are piezoelectric plates that have edge boundaries where wave energy is confined at the edge domain while it exponentially decays away from the edge.

Around a decade ago, Piliposian and Ghazaryan [9] studied localized bending waves at the free edge of a piezoelectric plate, addressing the possibility of edge wave existence

due to crack formation in a planar piezoelectric device. Recently, Nie et al. [10] have focused on an infinitely extended piezoelectric plate with an edge effect due to a bonded metallic strip. To the best of the authors' knowledge, nothing was published on addressing edge waves aside from these few papers. In this work, attention is focused on the practical case of when an edged piezoelectric plate is placed on an elastic substrate. Thus, a structure of interest composed of a piezoelectric plate that is infinitely extended in one direction and semi-infinitely extended in another direction, and resting on an elastic foundation, can be defined.

The interest in elastic edge waves propagating in semi-infinite plates started with the research of Konenkov [11] on the existence of edge waves along the edge of a Kirchhoff plate. The investigation of edge elastic waves has drawn increasing interest from researchers throughout the years [12–27].

The present paper is structured as follows: In Section 2, the equation of motion is described. In Section 3, the dispersion relation is obtained. In Section 4, numerical examples are provided on the effects of electrical loading, foundation stiffness and axial mechanical loading on the dispersion and attenuation characteristics. In Section 5, a conclusion is made.

## 2. Governing Relation

Consider a semi-infinite Kirchhoff piezoelectric plate (thick film) of thickness 2h resting on the Winkler–Fuss foundation as shown in Figure 1. The piezoelectric plate occupies the region specified by $(-\infty < X < \infty, 0 < Y < \infty, 0 < Z < 2h)$, with the foundation domain being given by $(-\infty < X < \infty, 0 < Y < \infty, 2h < Z < \infty)$.

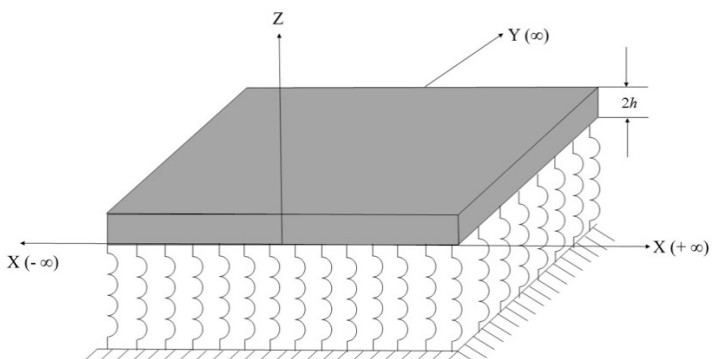

**Figure 1.** Elastic piezoelectric thick film on the Winkler−Fuss foundation.

Strain-Displacement Equations

$$\varepsilon_{xx} = -z\frac{\partial^2 w(x,y,t)}{\partial x^2} \varepsilon_{yy} = -z\frac{\partial^2 w(x,y,t)}{\partial y^2} \tag{1}$$

$$\varepsilon_{xy} = -z\frac{\partial^2 w(x,y,t)}{\partial x \partial y} \text{ or } \gamma_{xy} = -2z\frac{\partial^2 w(x,y,t)}{\partial x \partial y}. \tag{2}$$

Piezoelectric Constitutive Equations

$$\sigma_{xx} = c_{11}\varepsilon_{xx} + c_{12}\varepsilon_{yy} - e_{31}E_z \tag{3}$$

$$\sigma_{yy} = c_{22}\varepsilon_{yy} + c_{12}\varepsilon_{xx} - e_{31}E_z \tag{4}$$

$$\sigma_{xy} = 2c_{66}\varepsilon_{xy} = c_{66}\gamma_{xy}, \text{ where } \gamma_{xy} = 2\varepsilon_{xy} \tag{5}$$

$$D_z = e_{31}\varepsilon_{xx} + e_{31}\varepsilon_{yy} + \kappa_{33}E_z, \tag{6}$$

where $\sigma_{xx}, \sigma_{yy}, \sigma_{xy}$ are stress tensor components. $\varepsilon_{xx}, \varepsilon_{yy}, \varepsilon_{xy}$ are strain tensor components. $E_z$ is the electric field. $c_{11}, c_{22}, c_{12}, c_{66}$ are elastic constants. $e_{31}$ is the piezoelectric con-

stant, $W$ is the transverse displacement and $\kappa_{33}$ is the dielectric constant. $D_z$ is the electric displacement in the Z-dimension.

In the absence of electric charges, Gauss law is applied to the electric displacement, i.e., $\frac{\partial D_x}{\partial x} + \frac{\partial D_y}{\partial y} + \frac{\partial D_z}{\partial z} = 0$, $D_x, D_y$ are the electric displacements in the $X,Y$-dimensions, respectively. Assuming $D_x = D_y = 0$, i.e., the electric field is applied only across the thick film. Then, $\frac{\partial D_z}{\partial z} = 0$. The electric field is expressed as: $E_z = -\frac{\partial \Phi}{\partial z}$, where $\Phi$ is the electric potential. Then, substituting this in Equation (6) yields $D_z = e_{31}\varepsilon_{xx} + e_{31}\varepsilon_{yy} - \kappa_{33}\frac{\partial \Phi}{\partial z}$. Recall that $\frac{\partial D_z}{\partial z} = 0$. Then, $\frac{\partial}{\partial z}(e_{31}\varepsilon_{xx} + e_{31}\varepsilon_{yy}) - \kappa_{33}\frac{\partial^2 \Phi}{\partial z^2} = 0$, which can be written as $\frac{\partial^2 \Phi}{\partial z^2} = -\frac{e_{31}}{\kappa_{33}}\frac{\partial}{\partial z}[z\frac{\partial^2 w}{\partial x^2} + z\frac{\partial^2 w}{\partial y^2}]$ or, in terms of a first-order variation as $d\Phi = -\frac{e_{31}}{\kappa_{33}}[\frac{\partial^2 w}{\partial x^2} + \frac{\partial^2 w}{\partial y^2}]zdz$, on integrating both sides, we get $\int\limits_{-\frac{h}{2}}^{+\frac{h}{2}} d\Phi = -\frac{e_{31}}{\kappa_{33}}\int\limits_{-\frac{h}{2}}^{+\frac{h}{2}} [\frac{\partial^2 w}{\partial x^2} + \frac{\partial^2 w}{\partial y^2}]zdz$. Introducing $E_z = -\frac{\partial \Phi}{\partial z}$ gives $\int\limits_{-\frac{h}{2}}^{+\frac{h}{2}} E_z dz = -\frac{e_{31}}{\kappa_{33}}\int\limits_{-\frac{h}{2}}^{+\frac{h}{2}} [\frac{\partial^2 w}{\partial x^2} + \frac{\partial^2 w}{\partial y^2}]zdz$. Thus, $E_z = \frac{e_{31}}{\kappa_{33}}[\frac{\partial^2 w}{\partial x^2} + \frac{\partial^2 w}{\partial y^2}]z - \frac{V}{h}$.

Kirchhoff Plate Model

$$u(x,y,z,t) = u^0(x,y,t) - z\frac{\partial w(x,y,t)}{\partial x}, \tag{7}$$

$$v(x,y,z,t) = v^0(x,y,t) - z\frac{\partial w(x,y,t)}{\partial y}, \tag{8}$$

$$w(x,y,z,t) = w(x,y,t), \tag{9}$$

where $u^0(x,y,t)$ and $v^0(x,y,t)$ are the in-plane displacements of the mid-plane, which are assumed to be zero for the Kirchhoff plate. $w(x,y,t)$ is the out-of-plane (transverse) displacement of the plate.

Stress-Displacement Equations

$$\sigma_{xx} = -c_{11}z\frac{\partial^2 w}{\partial x^2} - c_{12}z\frac{\partial^2 w}{\partial y^2} + e_{31}\frac{\partial \Phi}{\partial z} \tag{10}$$

$$\sigma_{yy} = -c_{22}z\frac{\partial^2 w}{\partial y^2} - c_{12}\frac{\partial^2 w}{\partial x^2} + e_{31}\frac{\partial \Phi}{\partial z} \tag{11}$$

$$\sigma_{xy} = -2c_{66}z\frac{\partial^2 w}{\partial x \partial y}. \tag{12}$$

Shear Forces and Bending Moments

$$N_{ij} = \int\limits_{-\frac{h}{2}}^{+\frac{h}{2}} \sigma_{ij}dz \tag{13}$$

$$M_{ij} = \int\limits_{-\frac{h}{2}}^{+\frac{h}{2}} \sigma_{ij}zdz. \tag{14}$$

Deriving the Equations of Motion
Shear Forces

$$\frac{\partial N_{xx}}{\partial x} + \frac{\partial N_{xy}}{\partial y} = \rho h\frac{\partial^2 u^0}{\partial t^2} \tag{15}$$

$$\frac{\partial N_{xy}}{\partial x} + \frac{\partial N_{yy}}{\partial y} + = \rho h\frac{\partial^2 v^0}{\partial t^2} \tag{16}$$

Axial Force

$$\frac{\partial Q_x}{\partial x} + \frac{\partial Q_y}{\partial y} + q_z = \rho h \frac{\partial^2 w}{\partial t^2} \tag{17}$$

Bending Moments

$$\frac{\partial M_{xx}}{\partial x} + \frac{\partial M_{xy}}{\partial y} = \frac{1}{12}\rho h^3 \frac{\partial^2}{\partial t^2}\left(\frac{\partial w}{\partial x}\right) \tag{18}$$

$$\frac{\partial M_{xy}}{\partial x} + \frac{\partial M_{yy}}{\partial y} = \frac{1}{12}\rho h^3 \frac{\partial^2}{\partial t^2}\left(\frac{\partial w}{\partial y}\right), \tag{19}$$

where $\rho$ is the mass density, $q_z$ is the applied load. Then,

$$N_{xx}\left(\frac{\partial^2 w}{\partial x^2}\right) + 2N_{xy}\left(\frac{\partial^2 w}{\partial x \partial y}\right) + N_{yy}\left(\frac{\partial^2 w}{\partial y^2}\right) = q_z. \tag{20}$$

On solving, we get results subjected to forces and moments,

$$\begin{aligned}
\frac{\partial^2 M_{xx}}{\partial x^2} + \frac{\partial^2 M_{xy}}{\partial x \partial y} + \frac{\partial^2 M_{yy}}{\partial y^2} + \rho h \frac{\partial^2 w}{\partial t^2} &= \frac{1}{12}\rho h^3 \frac{\partial^2}{\partial t^2}\left(\frac{\partial^2 w}{\partial x^2} + \frac{\partial^2 w}{\partial y^2}\right) + \\
&\quad N_{xx}\left(\frac{\partial^2 w}{\partial x^2}\right) + 2N_{xy}\left(\frac{\partial^2 w}{\partial x \partial y}\right) + N_{yy}\left(\frac{\partial^2 w}{\partial y^2}\right).
\end{aligned} \tag{21}$$

The governing equation can be written in terms of transverse displacement, *W*, as:

$$\begin{aligned}
D_{11}\left(\frac{\partial^4 W}{\partial x^4}\right) + (D_{12} + 2D_{66})\left(2\frac{\partial^4 W}{\partial x^2 \partial y^2}\right) + D_{22}\left(\frac{\partial^4 W}{\partial y^4}\right) + 2\rho h\frac{\partial^2 W}{\partial t^2} + \beta W \\
= \frac{\rho h^3}{12}\frac{\partial^2}{\partial t^2}\left(\frac{\partial^2 W}{\partial x^2} + \frac{\partial^2 W}{\partial y^2}\right) + e_{13}V\left(\frac{\partial^2 W}{\partial x^2} + \frac{\partial^2 W}{\partial y^2}\right),
\end{aligned} \tag{22}$$

where *h* denotes half of the plate's thickness, $e_{31}$ is the piezoelectric constant, *V* is the applied voltage, $\rho$ is mass density, $\beta$ is the Winkler coefficient and the plate's flexural rigidities are:

$$D_{11} = \left(c_{11} + \frac{e_{31}^2}{\kappa_{33}}\right)\frac{h^3}{12}, \; D_{12} = \left(c_{12} + \frac{e_{31}^2}{\kappa_{33}}\right)\frac{h^3}{12}, \; D_{22} = \left(c_{22} + \frac{e_{31}^2}{\kappa_{33}}\right)\frac{h^3}{12} D_{66} = c_{66}\frac{h^3}{12}. \tag{23}$$

Note that the material parameters must satisfy the conditions of stability, ensuring positive definiteness of the strain energy. As the electrical parameters appear in coefficients of the governing Equation (22), only structural boundary conditions are needed to solve the problem. Thus, Equation (22) is solved with the following boundary conditions at the free edge $y = 0$:

$$\begin{aligned}
D_{12}\frac{\partial^2 W}{\partial x^2} + D_{22}\frac{\partial^2 W}{\partial y^2} &= 0, \\
(D_{12} + 4D_{66})\frac{\partial^3 W}{\partial x^2 \partial y} + D_{22}\frac{\partial^3 W}{\partial y^3} &= 0
\end{aligned} \tag{24}$$

## 3. Dispersion Analysis

Let us derive the dispersion relation for the bending edge of the wave. The solution of the plate Equation (22) is sought in the form of travelling harmonic wave as $w = We^{-i\omega t}$ then the equation of motion becomes:

$$D_{11}\left(\frac{\partial^4 W}{\partial x^4}\right) + 2(D_{12} + 2D_{66})\left(\frac{\partial^4 W}{\partial x^2 \partial y^2}\right) + D_{22}\left(\frac{\partial^4 W}{\partial y^4}\right) + \left(\frac{\omega^2 \rho h^3}{12} - e_{31}V\right)\left(\frac{\partial^2 W}{\partial x^2} + \frac{\partial^2 W}{\partial y^2}\right) + \left(\beta - 2\omega^2 \rho h\right)W = 0 \tag{25}$$

Let $D_{12} + 2D_{66} = H$, and then we get the following:

$$D_{11}\left(\frac{\partial^4 W}{\partial x^4}\right) + 2H\left(\frac{\partial^4 W}{\partial x^2 \partial y^2}\right) + D_{22}\left(\frac{\partial^4 W}{\partial y^4}\right) + \left(\frac{\omega^2 \rho h^3}{12} - e_{31}V\right)\left(\frac{\partial^2 W}{\partial x^2} + \frac{\partial^2 W}{\partial y^2}\right) + \left(\beta - 2\omega^2 \rho h\right)W = 0 \tag{26}$$

The governing equation accepts a solution for harmonic waves propagating in the +ve x− direction as:

$$W(x,y) = Ae^{ikx-k\lambda y},\tag{27}$$

where $k$ is the wavenumber and $\text{Re}(\lambda) > 0$ is a condition for ensuring decay or displacement in the Y direction.

Substituting Equation (27) into Equation (25), we arrive at the following bi-quadratic equation in $\lambda$:

$$12\,D_{11}k^4 + 12\beta - 24\,H\,k^4\lambda^2 + 12D_{22}k^4\lambda^4 - 12e_{31}k^2v\left(-1+\lambda^2\right) - 12\,h\rho\omega^2 - h^3k^2\rho\omega^2 + h^3k^2\rho\omega^2\lambda^2 = 0\tag{28}$$

On simplifying Equation (28), we get:

$$\lambda^4\left(12D_{22}k^4\right) + \lambda^2\left(24Hk^4 - 12e_{31}k^2v + h^3k^2\rho\omega^2\right) + \left(12D_{11}k^4 + 12\beta + 12e_{31}k^2v - 12h\rho\omega^2 - h^3k^2\rho\omega^2\right) = 0\tag{29}$$

After some transformations we deduce:

$$\lambda^4 + \lambda^2\left(\frac{h^3k^2\rho\omega^2 - 12e_{31}k^2v - 24Hk^4}{12D_{22}k^4}\right) + \left(\frac{12D_{11}k^4 + 12\beta + 12e_{31}k^2v - 12h\rho\omega^2 - h^3k^2\rho\omega^2}{12D_{22}k^4}\right) = 0\tag{30}$$

$$\lambda^4 - \lambda^2\left(\frac{2H}{D_{22}} + \frac{12e_{31}v - h^3\rho\omega^2}{12D_{22}k^2}\right) + \left(\frac{D_{11}}{D_{22}} + \frac{\beta - h\rho\omega^2}{D_{22}k^4} + \frac{12e_{31}v - h^3\rho\omega^2}{12D_{22}k^2}\right) = 0,\tag{31}$$

solving for the roots of Equation (31), which follow $\text{Re}(\lambda) > 0$,

$$\lambda_1 = \sqrt{\frac{H}{D_{22}} + \frac{e_{31}v}{2D_{22}k^2} - \frac{h^3\rho\omega^2}{24D_{22}k^2} - \frac{1}{24}\sqrt{-48\left(\frac{12D_{11}}{D_{22}} + \frac{12\,e_{31}v}{D_{22}k^2} + \frac{12\beta}{D_{22}k^4} - \frac{12h\rho\omega^2}{D_{22}k^4} - \frac{h^3\rho\omega^2}{D_{22}k^2}\right) + \left(\frac{-24H}{D_{22}} - \frac{12\,e_{31}v}{D_{22}k^2} + \frac{h^3\rho\omega^2}{D_{22}k^2}\right)^2}}\tag{32}$$

$$\lambda_2 = \sqrt{\frac{H}{D_{22}} + \frac{e_{31}v}{2D_{22}k^2} - \frac{h^3\rho\omega^2}{24D_{22}k^2} + \frac{1}{24}\sqrt{-48\left(\frac{12D_{11}}{D_{22}} + \frac{12\,e_{31}v}{D_{22}k^2} + \frac{12\beta}{D_{22}k^4} - \frac{12h\rho\omega^2}{D_{22}k^4} - \frac{h^3\rho\omega^2}{D_{22}k^2}\right) + \left(\frac{-24H}{D_{22}} - \frac{12\,e_{31}v}{D_{22}k^2} + \frac{h^3\rho\omega^2}{D_{22}k^2}\right)^2}}.\tag{33}$$

Simplifying the above equation, we get

$$\lambda_1 = \sqrt{\frac{H}{D_{22}} - \frac{(h^3\rho\omega^2 - 12e_{31}v)}{24D_{22}k^2} - \frac{1}{24}\sqrt{-48\left(\frac{12D_{11}}{D_{22}} - \frac{(h^3\rho\omega^2 - 12\,e_{31}v)}{D_{22}k^2} - \frac{12(h\rho\omega^2 - \beta)}{D_{22}k^4}\right) + \left(\frac{24H}{D_{22}} - \frac{(h^3\rho\omega^2 - 12\,e_{31}v)}{D_{22}k^2}\right)^2}}\tag{34}$$

$$\lambda_2 = \sqrt{\frac{H}{D_{22}} - \frac{(h^3\rho\omega^2 - 12e_{31}v)}{24D_{22}k^2} + \frac{1}{24}\sqrt{-48\left(\frac{12D_{11}}{D_{22}} - \frac{(h^3\rho\omega^2 - 12\,e_{31}v)}{D_{22}k^2} - \frac{12(h\rho\omega^2 - \beta)}{D_{22}k^4}\right) + \left(\frac{24H}{D_{22}} - \frac{(h^3\rho\omega^2 - 12\,e_{31}v)}{D_{22}k^2}\right)^2}}\tag{35}$$

Let

$$\alpha^2 = \frac{(h^3\rho\omega^2 - 12e_{31}v)}{D_{22}}\tag{36}$$

$$\gamma^4 = \frac{12(h\rho\omega^2 - \beta)}{D_{22}},\tag{37}$$

substituting Equations (34) and (35) in Equations (36) and (37), we get:

$$\lambda_1 = \sqrt{\frac{H}{D_{22}} - \frac{\alpha^2}{24k^2} - \frac{1}{24}\sqrt{-48\left(\frac{12D_{11}}{D_{22}} - \frac{\alpha^2}{k^2} - \frac{\gamma^4}{k^4}\right) + \left(\frac{24H}{D_{22}} - \frac{\alpha^2}{k^2}\right)^2}}\tag{38}$$

$$\lambda_2 = \sqrt{\frac{H}{D_{22}} - \frac{\alpha^2}{24k^2} + \frac{1}{24}\sqrt{-48\left(\frac{12D_{11}}{D_{22}} - \frac{\alpha^2}{k^2} - \frac{\gamma^4}{k^4}\right) + \left(\frac{24H}{D_{22}} - \frac{\alpha^2}{k^2}\right)^2}}.\tag{39}$$

Substituting these two roots $\lambda_1$ and $\lambda_2$ in Equation (40), we get:

$$W(x,y) = \sum_{j=1}^{2} C_j e^{ikx - k\lambda_j y}.\tag{40}$$

Substituting Equation (40) in the boundary conditions given in Equation (24) leads to

$$D_{12}{}^2 + 4D_{22}D_{66}\lambda_1\lambda_2 + D_{22}{}^2\lambda_1{}^2\lambda_2{}^2 + 4D_{12}D_{66} - D_{12}D_{22}\left(\lambda_1{}^2 + \lambda_2{}^2\right) = 0 \tag{41}$$

Substituting Equation (39) into Equation (41), and after solving, we get the following:

$$k^4\left(-12D_{12}{}^2 + 12D_{11}D_{22} + 2\,D_{22}D_{66}\sqrt{\frac{576D_{11}}{D_{22}} - \frac{48\alpha^2}{k^2} - \frac{48\gamma^4}{k^4}}\right) + k^2\left(D_{12}D_{22}\alpha^2 - D_{22}{}^2\alpha^2\right) = D_{22}{}^2\gamma^4. \tag{42}$$

Considering at the cutoff frequency $\alpha = \gamma = k$ in Equation (42), we get:

$$k^4\left(-12D_{12}{}^2 + 12D_{11}D_{22} + 2\,D_{22}D_{66}\sqrt{\frac{576D_{11}}{D_{22}} - 48 - 48}\right) + k^2\left(D_{12}D_{22}\alpha^2 - D_{22}{}^2\alpha^2\right) = D_{22}{}^2\gamma^4. \tag{43}$$

Substituting Equation (38) in Equation (43), and after simplification we get:

$$k^4\left(\frac{-12D_{12}{}^2}{D_{22}} + 12D_{11} + 2D_{66}\sqrt{\frac{576D_{11}}{D_{22}} - 96}\right) - k^2(D_{12} - D_{22})\frac{12e_{31}v}{D_{22}} + k^2\frac{(D_{12}-D_{22})}{D_{22}}h^3\rho\omega^2$$
$$= 12\left(h\rho\omega^2 - \beta\right). \tag{44}$$

Simplifying Equation (44), we get:

$$D_{11}k^4\left(\frac{-D_{12}{}^2}{D_{11}D_{22}} + 1 + \frac{D_{66}}{6D_{11}}\sqrt{\frac{576D_{11}}{D_{22}} - 96}\right) - k^2\left(\frac{D_{12} - D_{22}}{D_{22}}\right)e_{31}v + k^2\frac{(D_{12} - D_{22})}{12D_{22}}h^3\rho\omega^2 = h\rho\omega^2 - \beta. \tag{45}$$

The rotational inertia and the piezoelectric terms in Equation (45) are given respectively as follows:

$$k^2\frac{(D_{12} - D_{22})}{12D_{22}}h^3\rho\omega^2\,\&\,k^2\left(\frac{D_{12} - D_{22}}{D_{22}}\right)e_{31}v. \tag{46}$$

Here, if we neglect the rotational inertia and the piezoelectric terms given in Equation (46), Equation (45) becomes:

$$D_{11}k^4\left(\frac{-D_{12}{}^2}{D_{11}D_{22}} + 1 + \frac{D_{66}}{6D_{11}}\sqrt{\frac{576D_{11}}{D_{22}} - 96}\right) = h\rho\omega^2 - \beta. \tag{47}$$

The dispersion relation may be re-casted as:

$$D_{11}k^4\delta^4 = h\rho\omega^2 - \beta, \tag{48}$$

where

$$\delta^4 = \left(\frac{-D_{12}{}^2}{D_{11}D_{22}} + 1 + \frac{D_{66}}{6D_{11}}\sqrt{\frac{576D_{11}}{D_{22}} - 96}\right) \tag{49}$$

$$\frac{D_{11}k^4\delta^4}{\beta} = \frac{h\rho\omega^2}{\beta} - 1 \tag{50}$$

The dispersion relation can be represented in dimensionless form as:

$$K^4 = \Omega^4 - 1 \tag{51}$$

This coincides with the results of Althobaiti et al. [26].

Therefore, the following equation includes the rotational inertia as well as the piezoelectric terms, as also given in Equation (45):

$$D_{11}k^4\left(\frac{-D_{12}{}^2}{D_{11}D_{22}} + 1 + \frac{D_{66}}{6D_{11}}\sqrt{\frac{576D_{11}}{D_{22}} - 96}\right) - k^2\left(\frac{D_{12} - D_{22}}{D_{22}}\right)e_{31}v + k^2\frac{(D_{12} - D_{22})}{12D_{22}}h^3\rho\omega^2 = h\rho\omega^2 - \beta. \tag{52}$$

## 4. Numerical Results and Discussions

This section presents several numerical examples to demonstrate the dispersion curves of bending edge waves' propagation along the free edge of a semi-infinite piezoelectric plate resting on a Winkler–Fuss foundation. When a local minimum of the phase velocity coincides with the value of the group velocity, it results in the critical speed of the applied load in the form of bending edge waves.

Here, let us initiate the discussion of the results by considering the work presented by Althobaiti et al. [26] as the base and then adding the rotational inertia term to it, followed by the piezoelectric terms. In order to facilitate comparison of the dispersion curves obtained by including the rotational inertia and the piezoelectric terms with that presented in [26], let us restrict the work in [26] as level-1, including rotatory inertia terms as level-2, adding piezoelectric terms to level-1 as level-3 and finally adding both the rotary inertia and the piezoelectric terms to the level-1 as level-4; the dispersion equation for level-4 is given in equation (27).

The density of the piezoelectric plate is considered as $\rho = 7500 \, \text{kg/m}^3$, the half thickness of the plate as 0.1 mm, and the Winkler constant as $\beta = 10^5 \, \text{N/m/m}$. The piezoelectric material is of the type PZT−H, whose relevant constants are $c_{11} = 102$ GPa, $c_{12} = 31$ GPa, $c_{66} = 35.5$ GPa, $e_{31} = -17.05 \, \text{C/m}^2$, $K_{33} = 1.76 \times 10^{-8} \, \text{C/Vm}$.

The dispersion curves from level-1 and level-2 are plotted in Figure 2. It could be observed that the dispersion curves from these two levels superimpose each other with the same critical velocity of 11.01 m/s at the circular frequency of 517.97 rad/s. Therefore, it is evident that the rotary inertial terms have a negligible effect on the critical velocity.

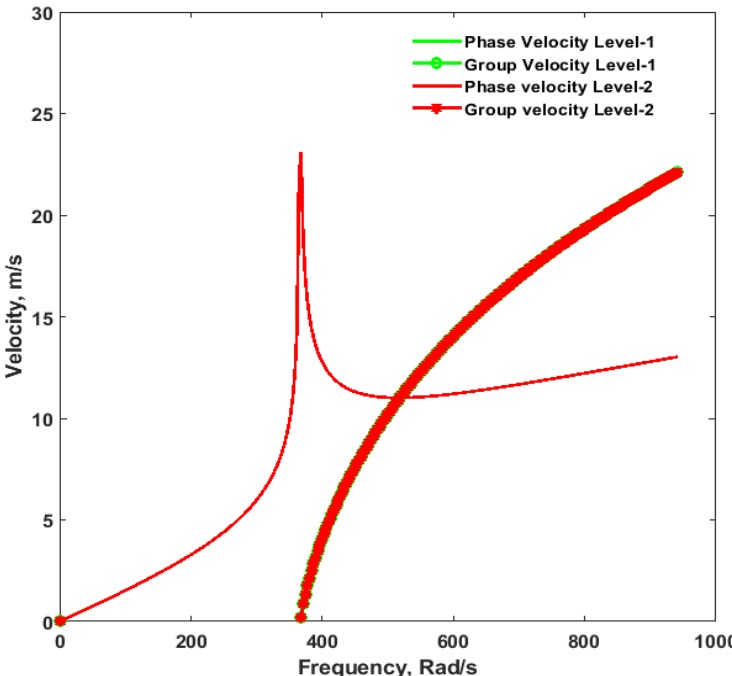

**Figure 2.** Comparison of Dispersion Curves for Level-1 and Level-2.

Let us compare the dispersion curves for levels 1 and 3, where piezoelectric terms were added to level-1, i.e., in Equation (47). Dispersion curves are given in Figure 3, where it can be seen that the critical velocity is lowered by adding the piezoelectric effect to level-1. The critical velocity for level-3 is obtained as 9.694 m/s at the circular frequency of 456.281 rad/s, whereas the critical velocity for level-1 was 11.01 m/s at 517.97 rad/s.

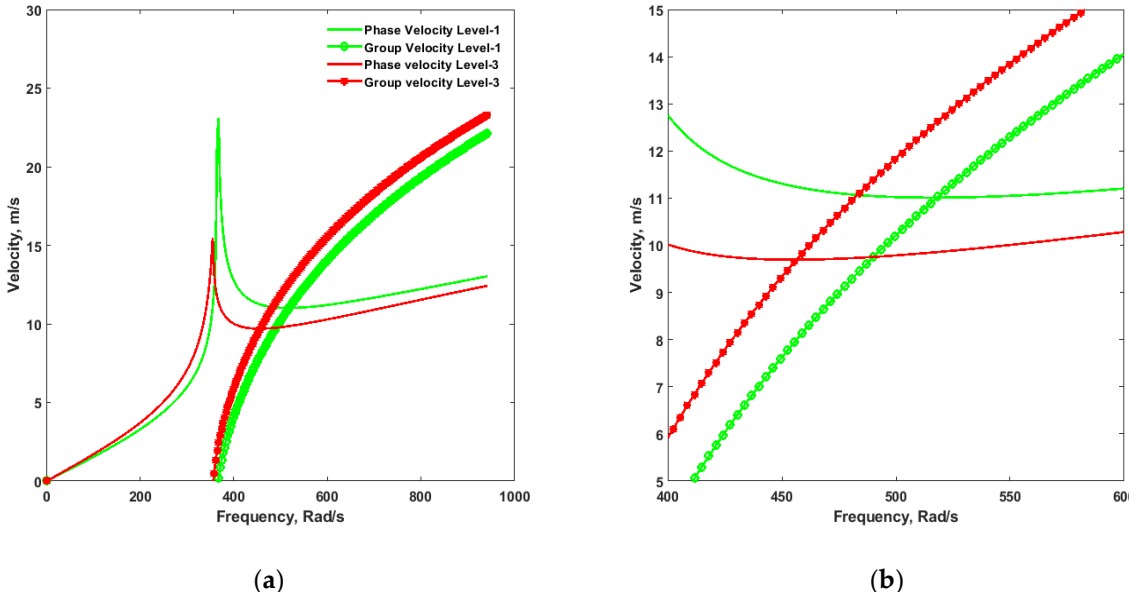

(**a**)　　　　　　　　　　　　　　　　(**b**)

**Figure 3.** (**a**) Comparison of Dispersion Curves for Level-1 and Level-3, (**b**) Close View of (**a**).

Figure 4 shows the dispersion curves for level-1, which are plotted against the dispersion curves for the case where both the rotary inertia and the piezoelectric terms are added to it. The critical velocity for level-4 was obtained as 9.694 m/s at the frequency of 456.281 rad/s vis-à-vis the critical velocity of 11.01 m/s at the frequency of 517.97 rad/s for level-1. As observed from both Figures 3 and 4, the critical velocities were the same, and the rotary inertia effects were negligible.

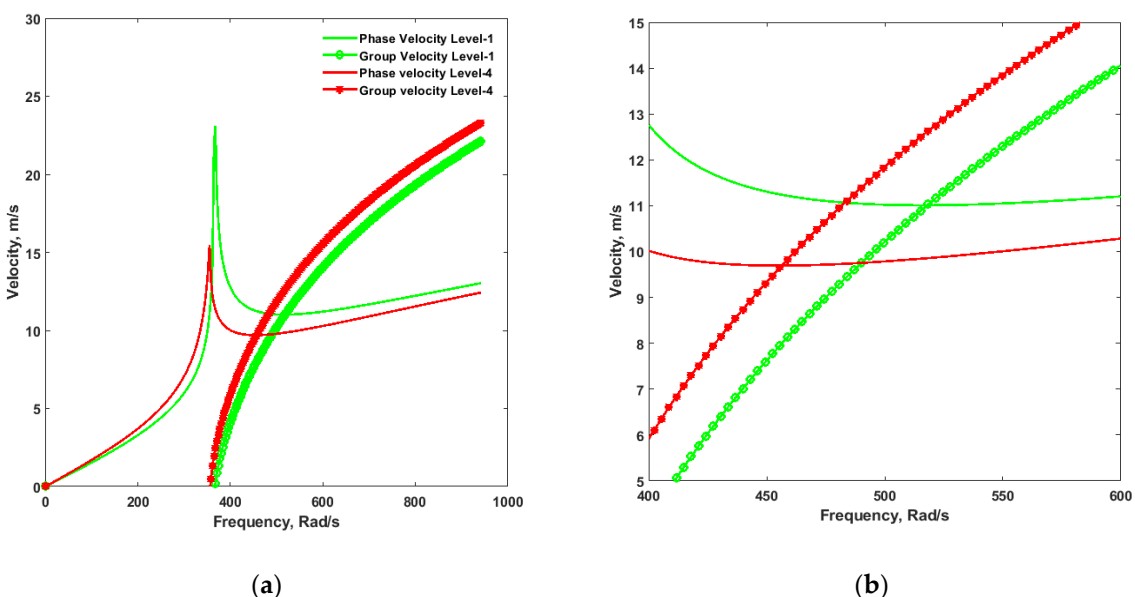

(**a**)　　　　　　　　　　　　　　　　(**b**)

**Figure 4.** (**a**) Comparison of Dispersion Curves for Level-1 and Level-4, (**b**) Close View of (**a**).

Let us now consider the impact of changing the voltages on level-4, where both the inertial and the piezoelectric effects are considered, and study the nature of critical velocity. Figure 5a shows the dispersion curves for level-4 by varying the voltage at a step of 5 V from 0 to 5 V.

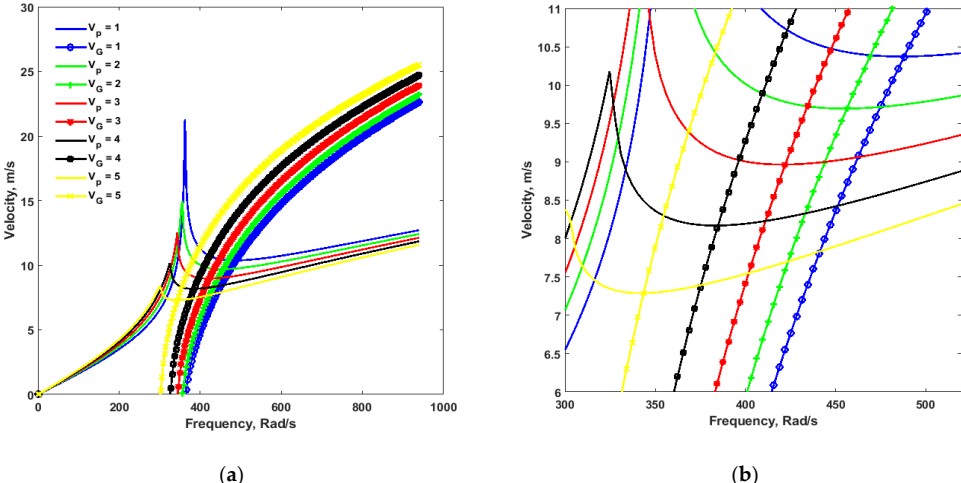

(**a**)                                   (**b**)

**Figure 5.** (**a**) Dispersion Curves for Level-4 at Varying Voltages. (**b**) Close View of Figure 5a.

It can be seen from Figure 5b, which is the close-up view of Figure 5, that the critical velocity decreases with an increase in the voltage from 1 V to 5 V. The critical velocity is 10.37 m/s at the frequency of 488.10 rad/s at 1 V. Then it decreases to 9.69 m/s at a frequency of 456.20 rad/s at 2 V. It further decreases to 8.960 m/s at a frequency of 422.04 rad/s at 3 V. It further decreases to 8.16 m/s at a frequency of 384.70 rad/s at 4 V. Finally, it decreases to 7.28 m/s at a frequency of 343.40 rad/s at 5 V.

The discussion of the dispersion curves in the above presented figures was about the value of the Winkler's constant at $10^5$ N/m/m. Now, let us consider the effect of increasing the Winkler's constant from $10^5$ N/m/m to $10^6$ N/m/m and study the variation of critical velocity by changing the voltage from 0 V to 5 V at a step of 1 V.

Figure 6a shows the dispersion curves for the bending edge waves in a piezoelectric Kirchhoff plate resting on a Winkler's foundation with the stiffness of $\beta = 10^6$ N/m/m, and the voltage is varied from 1 V to 5 V. It is evident from Figure 6b, the close-up view of Figure 6a, that the critical velocity decreases with the increase of the voltage. It could also be observed that the numerical values of the critical velocities with the Winkler's constant $\beta = 10^6$ N/m/m are larger than the critical values obtained with $\beta = 10^5$ N/m/m.

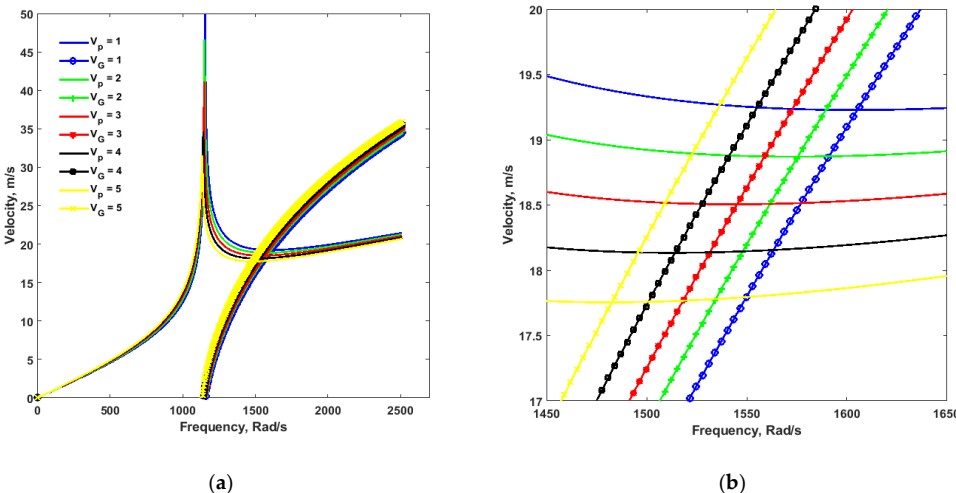

(**a**)                                   (**b**)

**Figure 6.** (**a**) Dispersion Curves of Level-4 at Varying Voltages for $\beta = 10^6$ N/m/m. (**b**) Close View of Figure 6a.

It is clear from Figure 6b that the critical velocity is 19.20 m/s at frequency 1605.20 rad/s at 1 V. It decreases to 18.87 m/s at the frequency 1575.40 rad/s at 2 V. Further, it decreases

to 18.50 m/s at a frequency of 1545.10 rad/s at 3 V. Then, it drops to 18.1 m/s at a frequency of 1514.09 rad/s at 4 V. Finally, it decreases to 17.70 m/s at a frequency of 1482.40 rad/s at 5 V.

## 5. Conclusions

The present study considers the propagation of flexural edge waves on a thick piezo-electric film (thin piezoelectric plate) supported by a Winkler–Fuss foundation. The dispersion relation has been obtained. The variation of critical velocity by including the rotary inertia and the piezoelectric terms in the dispersion equation obtained in Althobaiti et al. [27] is studied. It was observed that the rotary inertia has a negligible influence on the dispersion curves. The influence on critical velocity for the dispersion curves including both the effects of rotary inertia and the piezoelectricity has also been studied for varying voltages, and it was found that the critical velocity decreases with an increase in the voltage and increases with increases in the Winkler's constant.

**Author Contributions:** Validation, S.A.; writing—review and editing, S.A., M.A.H.; supervision, M.A.H. All authors have read and agreed to the published version of the manuscript.

**Funding:** Research support offered by the Deanship of Scientific Research, Taif University, Saudi Arabia [1-441-89].

**Institutional Review Board Statement:** Not applicable.

**Informed Consent Statement:** Not applicable.

**Data Availability Statement:** Not applicable.

**Conflicts of Interest:** The authors declare no conflict of interest.

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
