# Peer review of "Flexural Edge Waves in a Thick Piezoelectric Film Resting on a Winkler Foundation"

_crystals, doi:10.3390/cryst12050640_

Round 1

Reviewer 1 Report

The authors study theoretically the bending wave propagation along the edge of a semi-infinite thin plate.

I think the paper should be rejected because of the error in Eq. (6). Namely, the contribution of e_{33}\varepsilon_{zz} is missed in it and subsequent ones.  

Author Response

The authors are very grateful to the referee for the helpful comments. We are considering a plane stress problem, we will provide some reference to justify our equation.

Reviewer 2 Report

The paper is devoted to the study of flexural edge waves in a thin piezoelectric plate resting on a Winkler foundation. The paper in my opinion needs very significant revision. After that it can be published. In my opinion, there are a lot of minor flaws and typos in the text of the manuscript. For example some of them,

  1. Line 68 shows the foundation region is (-∞<X <∞, 0< Y <∞, 2h< Z < ∞).  There seems to be no correspondence with Figure 1.
  2. It is stated that the thickness of the plate is 2h at the beginning of the manuscript and in Figure 1,  but in the other part of the paper the thickness is denoted as h  ("where h denotes the plate's thickness," line 103).  The integration in the formulas is chosen from -h/2 to h/2.
  3.  Why is the vector Q in formula (17) not defined in the text of the manuscript?
  4.  There seems to be confusion in lines 102,114 and formulas (21), (22), (25) with notation of transverse displacement.
  5. It is necessary to unify the size of all formulas. There is too big a contrast in sizes.
  6.  For formulas (11)-(21) more text explanations are needed.
  7.  What is the unit of measure "N/m/m" for Winkler constant ? 

There are many other small incomprehensible points in the text that need to be addressed.

It is desirable to additionally compare the analytical model with other approaches, such as the finite-element method in COMSOL. As it was done in another paper of the authors. Why did not the authors refer to their earlier interesting paper on very similar topics in WM journal?

Siddiqui, M. A., & Hawwa, M. A. (2021). Flexural edge waves in a Kirchhoff plate carrying periodic edge resonators and resting on a Winkler foundation. Wave Motion, 103, 102720.

The number of paper on wave propagation in periodic edge structures is not very numerous at present. Therefore, it is quite appropriate to refer to recent works on similar topics.

Nazarov, S. A. (2021). Trapping of Waves in Semiinfinite Kirchhoff Plate with Periodically Damaged Edge. Journal of Mathematical Sciences, 257(5), 684-704.

Althobaiti, S. N., Nikonov, A., & Prikazchikov, D. (2021). Explicit model for bending edge wave on an elastic orthotropic plate supported by the Winkler–Fuss foundation. Journal of Mechanics of Materials and Structures, 16(4), 543-554.

Nedospasov, I. A., Pupyrev, P. D., Bechler, N., Tham, J., Kuznetsova, I. E., & Mayer, A. P. (2022). Guided acoustic waves at periodically structured edges: Linear modes and nonlinear generation of Lamb and surface waves. Journal of Sound and Vibration, 116854.

Author Response

The authors are very grateful to the referee for the helpful comments. All of those were thoughtfully addressed. The summary of implemented changes is attached 

Reviewer 3 Report

The authors investigate the propagation of flexural edge waves along the edge of a thin semi-infinite piezoelectric film attached to a foundation. The dispersion relation has been derived using two simplifying assumptions that Kirchhoff-type piezoelectric film is considered, and that substrate can be substituted by Winkler-Fuss foundation.  Though two simplifications are assumed, the problem is of certain scientific interest. Besides, the results of the numerical analysis showing the influence of the waveguide properties (the Winkler’s constant, voltage) on the dispersion properties. Therefore, I recommend the paper for publication after a certain revision.

  1. The statement of the problem is not clearly described. Specifically, boundary conditions for electric potential are not straightforwardly written (the relation given in line 91 should be given more carefully) and some of the introduced values have not been described in the text.
  2. 0.5 mm thickness plate is not so thin, subsequently comments related to the accuracy of assumptions should be given in the text.
  3. The limitations caused of the assumptions made should be described in the sense of the expected divergence compared with more complex models is suggested.

Author Response

Dear Respected Editor,

We would like to thank the anonymous reviewer for his/her valuable comments. Following are our response for these comments:

  • The relations given at and beyond line 91 are made clear and highlighted. As for the electrical boundary conditions, the problem considered include electrical (piezoelectric) elements in the coefficients of the governing equation (22). Therefore, only structural boundary conditions are needed.

  • We fully agree with the referee that our plate should be classified as a thick piezoelectric film. Hence, the calculations were repeated for a piezoelectric plate having a thickness of 0.1 mm, which is within the classification of thick films / crystals. As the plate becomes thinner, the electric field becomes larger. Therefore, the maximum voltage used was 5 volts.

  • We referred to the fact that we have used a “reduced” electro-mechanical model, where the equation electric equation (conservation of charge) did not appear as independent equation in the model considered. The reduced model should work fine if the semiconducting nature of the piezoelectric material is neglected.

Again, we appreciate the referee’s comments and we thank you for the efforts you spent on our manuscript.

Saad Althobaiti and Muhammad Hawwa      

Round 2

Reviewer 2 Report

The comments have been all addressed by the authors. 

Author Response

Again, we appreciate the referee’s comments and we thank you for the efforts you spent on our manuscript.